# A Sense of Scarcity Enhances the Above-Average Effect in Social Comparison

**DOI:** 10.3390/bs13100826

**Published:** 2023-10-08

**Authors:** Xiaoyan Wang, Lan Jiao

**Affiliations:** 1College of Psychology, Sichuan Normal University, Chengdu 610000, China; xywang@sicnu.edu.cn; 2College of Teacher Education, Ningbo University, Ningbo 315000, China

**Keywords:** sense of scarcity, social comparison, above-average effect, mental health

## Abstract

Scarcity refers to a state in which an individual’s resources do not satisfy his/her needs. A sense of scarcity evokes negative emotions. A fundamental strategy for coping with this negative threat is for people to emphasize the desirability of their personal traits. In this study, a 2 (sense of scarcity: high or low) × 2 (valence: positive or negative) mixed-design experiment was conducted to examine whether and how a sense of scarcity affected one’s self-evaluation. Participants were assigned randomly to a high- or low-scarcity group. The chances of assistance rendered to an individual during a word puzzle task were manipulated to induce a high or low sense of scarcity. Then, participants were asked to make positive and negative trait judgments of themselves compared with their average peers. The results showed that people judged their personalities to be more desirable (i.e., more positive and less negative traits) than their average peers, manifesting the above-average effect. More importantly, people with a high sense of scarcity manifested a greater above-average effect than those with a low sense of scarcity. This study suggests that people could highlight their positive aspects to cope with predicaments in social life.

## 1. Introduction

In modern society, people often experience scarcity. Scarcity refers to a state in which an individual’s resources do not satisfy his/her needs [1]. Previous studies have documented that people value and make the best use of the scarce resources they have, enhancing their creativity in product use [1,2]. However, a sense of scarcity exerts many negative effects, such as overborrowing, impaired cognitive function, and an increased time discount rate in intertemporal decision-making [1,3,4]. More importantly, a sense of scarcity is a negative affection, and it might damage one’s physical and mental health. Under such circumstances, people might take actions to serve as adaption functions. Thus, the present study mainly focused on exploring whether and how people regulate themselves to adapt to scarcity situations. It is important for us to better understand the self-regulation mechanisms of individuals under scarce circumstances, which may support the elimination of the negative effects of scarcity.

A sense of scarcity might be from a lack of food, money, time, or resources [1,3,5]. For example, Nelson and Morrison [5] operationalized resource scarcity through feelings of financial and caloric dissatisfaction to explore its effect on preference for potential partners. Results showed that men who feel either poor or hungry prefer heavier women than men who feel rich or full. In a study by Shah, Mullainathan, and Shafir [1], scarcity was manipulated by budgeting participants’ chances to guess letters in word puzzles. High sense of scarcity participants had fewer chances than those with a low sense of scarcity. These researchers found that people with a high sense of scarcity engaged more deeply in some problems while neglecting others, such as overborrowing. Furthermore, Mani et al. [6] directly distinguished participants into “poor” or “rich” categories based on their financial state. They found that the same farmer showed diminished cognitive performance when poor as compared with when rich. A recent study also showed that people with financial resource scarcity tend to avoid information about environmental perils. Remaining uninformed is short-sighted and problematic because it may increase people’s vulnerability to damage from these environmental threats. These results suggested that a sense of scarcity made people pursue immediate gains while ignoring long-term benefits. Furthermore, a sense of scarcity from a lack of money, food, or resources could also cause many negative psychological effects, such as negative emotions and doubt about his/her self-worth or ability, undercutting meaning in life and threatening self-esteem [7,8,9]. Fortunately, humans have a great capacity to regulate themselves to adapt to hard times.

The self is one of the core characteristics of human experience. The way people undertake self-evaluation plays an important role in survival and maintaining mental health [10,11,12,13]. Numerous studies have found that people tend to evaluate themselves positively [14,15,16]. For example, in social comparisons, most people judge their personality to be more desirable (i.e., having more positive and less negative traits) than their average peers, which is termed the above-average effect [17,18]. A motivational explanation has claimed that the above-average effect derives from the desire to enhance the positivity or diminish the negativity of one’s self-concept. It is a way for individuals to protect their fragile egos from the blows of reality [17]. Under the circumstances of scarcity, people might experience negative emotions, and their self-worth and self-esteem might be threatened. Individuals with a sense of scarcity may not be satisfied with their current negative situations. This dissatisfaction may prompt them to make self-regulation strategies to cope with their predicament. Hughes and Beer [19] found that self-evaluations made in response to social threats and negative emotions significantly increased the above-average effect of participants. Emphasizing the desirability of their personal characteristics was considered a fundamental way that people protect themselves from challenging times. Additionally, the above-average effect was considered one of the heuristic responses that allows people to maintain a positive self-view. Inhibition of the heuristic above-average effect usually requires additional cognitive resources [15,20,21,22,23]. Thus, we speculated that people with a high sense of scarcity might engage in self-protection strategies by emphasizing their desirability in social comparison. Additionally, we hypothesized that they would respond faster to make such a heuristic response.

In summary, although numerous studies have explored the effect of a sense of scarcity on human behavior, little evidence has directly clarified how people regulate themselves to adapt to scarce situations. Based on a previous study [1], the chances of assistance rendered to an individual during a word puzzle task were manipulated to induce a high or low sense of scarcity. The above-average effect was measured in a social comparison task where participants were asked to make trait judgments compared with their average peers [18]. We hypothesized that compared with those with a low sense of scarcity, individuals with a high sense of scarcity would respond faster and manifest a greater above-average effect to maintain a positive self-view in social comparison.

## 2. Materials and Methods

### 2.1. Study Design

In the present experiment, we used a 2 (sense of scarcity: high or low) × 2 (valence: positive or negative) mixed design, with participants’ sense of scarcity as the between-subject factor and the valence as the with-subject factor. Participants’ ratings on positive and negative personality traits in a social comparison task were used as the main outcome measures.

### 2.2. Participants

Participants were recruited via the research participation board of Sichuan Normal University (SICNU). To be included in the study, participants needed to be college students with normal or corrected normal vision, at least 17 years old, and speak Chinese. The experiment was conducted in the laboratory of SICNU. Participants signed an informed consent form before completing the experimental tasks and received monetary remuneration of 10 CNY upon completion of the experiment. This experiment was approved by the ethics committee of Sichuan Normal University. Further information about the inclusion and exclusion of participants and the demographics of the final sample can be found in the Participants sub-section of the Results section.

### 2.3. Materials

#### 2.3.1. Word Puzzles Task

Referring to the previous study [1], participants’ senses of scarcity were manipulated by budgeting participants’ chances of assistance rendered in a word puzzle task. The word puzzle task consisted of 10 items of moderate difficulty, which were selected from the previous study [24]. For example, “puzzle: person in the mirror; answer: enter” (the Chinese character for person is “人” and the character for enter is “入”). The goal of the participant was to guess all answers, and the number of correct guesses was related to the compensation they received for participating. The participant could try to answer alone or seek assistance from the experimenter, who would provide a character cue as a hint. For the high sense of scarcity group, participants were informed that they could have 3 hints for the entire word puzzle task. For the low sense of scarcity group, participants were informed that they could have 10 hints for the entire word puzzle task. The number of requested hints and the number of correct answers for every participant were recorded.

#### 2.3.2. Social Comparison Task

The social comparison task has been widely used in measuring one’s above-average effect in previous studies [18,19]. This task includes 100 personality items (50 each of positive and negative terms). There were significant differences in valence between positive and negative terms, *t* (98) = 57.59, *p* < 0.001, but not in familiarity, *t* (98) = 0.44, *p* = 0.66. As shown in Figure 1, each trial was initiated by an 800 ms presentation of a black fixation cross on a white background. Then, a 500 ms blank was followed by a social comparison question. Participants were asked to compare themselves with their average peers using a 5-point scale (–2 = much less than the average peer group, 0 = about the same as the average peer group, 2 = much more than the average peer group). The item appeared for an unlimited time and disappeared after the participant selected a button to make a response. Participant’s rating and response time for each item was recorded separately. All the trials were presented in a random order. Before the formal experiment, each participant completed 10 items as practice.

### 2.4. Procedure

The experiment was conducted by a graduate student who has received professional training in psychological research for two years or an assistant professor in the university who has published several research papers in academic journals. In the experiment, participants were measured individually and face-to-face with the experimenter in the laboratory.

Each participant was asked firstly to complete the word puzzle task to activate his/her high or low sense of scarcity. After the word puzzle task, each participant was asked to answer questions on a 7-point scale to complete the scarcity manipulation test, for example: “Do you feel that the surrounding environment could enable you to focus on solving the problem?” “Do you feel that there are too many or too few questions?” “Do you think the number of hints in the task is sufficient?” Of these, question 3 was the critical question to measure the effectiveness of scarcity priming in the participant. After that, the participant was asked to complete the social comparison task to measure the above-average effect and response time. Finally, participants were debriefed on the study.

### 2.5. Outcomes

The primary outcome of the present experiment was the difference in the above-average effect between participants with high or low senses of scarcity. After each participant completed the social comparison task, his/her ratings for positive and negative personality traits were calculated separately to measure the above-average effect. Specifically, the ratings directly indicated the above-average effect in positive traits. For negative traits, the ratings needed to be reversed to indicate the above-average effect. The higher the score, the stronger the individual’s above-average effect in social comparison.

In addition, participants’ ratings for the question “Do you think the number of hints in the task is sufficient?” were calculated to check the manipulation of the sense of scarcity.

### 2.6. Sample Size

To determine an appropriate sample size to test our hypothesis, we conducted an a priori power analysis for a 2 (sense of scarcity: high or low) × 2 (valence: positive or negative) repeated measured ANOVA using G*Power Version 3.1 [25,26]. Expecting a medium effect size (*f* = 0.25) and aiming for high statistical power (1 − β = 0.95 with α = 0.05), the a priori power analysis indicated that a total sample of 54 participants would be appropriate. Oversampling slightly, we recruited a total of 69 participants.

These 69 college students (58 females) were randomly allocated into high- or low-scarcity groups. Specifically, all participants were assigned a number according to their sequence of signing up for participation. Participants with odd numbers were assigned to the high sense of scarcity group, and those with even numbers were assigned to the low sense of scarcity group.

### 2.7. Statistical Methods

Firstly, independent sample *t*-tests were used to examine the difference in age and educational years between the high and low sense of scarcity groups. Secondly, independent sample *t*-tests were used to measure the effectiveness of manipulation on the sense of scarcity. Thirdly, one-sample *t*-tests were used to check whether there was an above-average effect in the social comparison task. At last, a 2 (sense of scarcity: high or low) × 2 (valence: positive or negative) repeated measures ANOVA was conducted to examine the effect of the sense of scarcity on the above-average effect.

We used the CONSORT checklist when writing our report [27].

## 3. Results

### 3.1. Participants

As the data of 1 participant were lower by more than 3 standard deviations, this participant was removed, and the data of the remaining 68 participants were included in the final statistical analysis, with 34 each in the high- and low-scarcity groups. There was no significant age difference between the groups with high (26 females, 17–25 years old, *M*= 19.38, *SD* = 1.02) and low (32 females, 18–26 years old, *M_age_* = 19.26, *SD* = 0.96) senses of scarcity, *t* (66) = 0.29, *p* = 0.83. Additionally, the education years between the two groups were not significant (high sense of scarcity group: *M* = 14.00, *SD* = 0.82; low sense of scarcity group: *M* = 13.85, *SD* = 0.78), *t* (66) = 0.79, *p* = 0.45. These results revealed that participants in the two groups matched well in age and education years. The effect of a sense of scarcity on the above-average effect might not be modulated by these factors.

### 3.2. Manipulation Check

An independent sample *t*-test found that scarcity perception in the high-scarcity group (*M* = 3.18, *SD* = 1.91) was marginally significantly higher than in the low-scarcity group (*M* = 2.35, *SD* = 1.63), *t* (66) = 1.91, *p* = 0.06, Cohen’s *d* = 0.48). In addition, an independent sample *t*-test of the number of requested hints for the two groups found that the high-scarcity group (*M* = 2.36, *SD* = 1.08) used fewer hints than the low-scarcity group (*M* = 4.97, *SD* = 2.88), *t* (39.82) = 4.80, *p* < 0.001, Cohen’s *d* = 1.20). These results showed that the manipulation of the sense of scarcity in this study was effective.

Additionally, there were 100 trials randomly presented in the social comparison task. Participants spent a relatively long time answering all rating questions, and fatigue or disinterest might have occurred at some point. A paired-sample *t*-test was used to check these possible effects. The results showed that there was no difference in ratings between the first 50% of responses and the final 50% of responses (the first 50% of responses: *M* = 0.48, *SD* = 0.41; the final 50% of responses: *M* = 0.49, *SE* = 0.45, *t* (67) = 0.46, *p* = 0.67). These results revealed that the above-average effect measured in the present study was stable and reliable.

### 3.3. Effect of Sense of Scarcity on the Above-Average Effect

As shown in Table 1, the ratings and response times of participants with a high or low sense of scarcity on positive and negative traits were calculated. The ratings on negative traits were reversed scored. One-sample *t*-tests showed that participants judged their personalities to be more positive (*t* (67) = 5.54, *p* < 0.001, Cohen’s *d* = 0.68) and less negative (*t* (67) = 10.62, *p* < 0.001, Cohen’s *d* =1.30) than their average peers. The results revealed that individuals manifested the above-average effect in social comparison.

General linear models were used to further examine how a sense of scarcity affected one’s above-average effect in social comparison. A 2 (sense of scarcity: high or low) × 2 (valence: positive or negative) repeated measures ANOVA on the ratings showed main effects of sense of scarcity [*F* (1, 66) = 4.32, *p* = 0.05, $\eta_{p}^{2}$ = 0.06] and valence [*F* (1, 66) = 55.39, *p* = 0.001, $\eta_{p}^{2}$ = 0.46]. The interaction between sense of scarcity and valence was not significant, *F* (1, 66) = 2.03, *p* = 0.16. Pairwise comparisons showed that participants with a high sense of scarcity had higher ratings than those with a low sense of scarcity in the social comparison task. Participants also showed lower ratings on evaluating positive traits than on evaluating negative traits. That is, participants with a high sense of scarcity manifested a greater above-average effect than those with a low sense of scarcity. Furthermore, participants showed a lower above-average effect when evaluating positive traits than when evaluating negative traits.

Additionally, a repeated-measures ANOVA on participants’ RT in the social comparison task showed that there was no main effect of sense of scarcity, *F* (1, 66) = 0.84, *p* = 0.36, valence, *F* (1, 66) = 1.64, *p* = 0.21, or an interaction between scarcity and valence, *F* (1, 66) = 1.26, *p* = 0.27. These results revealed that a sense of scarcity did not affect participants’ response time in the social comparison task.

In the experiment, the chances of assistance rendered to an individual during a word puzzle task were manipulated to examine the effect of a sense of scarcity on the above-average effect. The results showed that the group with a high sense of scarcity had a stronger above-average effect than the group with a low sense of scarcity. Additionally, the above-average effect of participants regarding positive traits was significantly lower than that regarding negative traits.

## 4. Discussion

The present study manipulated a sense of scarcity in individuals to examine the causal relationship between a sense of scarcity and the above-average effect in social comparison. The results showed that participants with a high sense of scarcity reported having more positive and fewer negative traits than did participants with a low sense of scarcity—that is, a sense of scarcity enhanced the above-average effect in social comparison. There was no significant difference in reaction time between the two groups in the social comparison task. Additionally, we also found that people manifested greater above-average effects in negative traits than in positive events.

In social comparison, people with a high sense of scarcity claimed more positive and less negative traits than those with a low sense of scarcity, manifesting a greater above-average effect. The above-average effect originates from an individual’s need to form and maintain a positive self-concept, resulting in self-enhancement and self-protection. Self-enhancement helps an individual form a positive self-concept or self-image, while self-protection is an emergency response system that operates when the self-concept or self-image is under threat [28]. For people with a sense of scarcity, emphasizing the desirability of their characteristics might be a good way to protect themselves from a negative, scarce reality. However, we did not find a response time difference between the two groups. One of the possibilities was that people with a high sense of scarcity did not consume more cognitive resources in the word puzzle task. People had equal cognitive resources to make their self-evaluations in the following social comparison task. The self-management model of resource scarcity states that people usually have two different response pathways when faced with resource scarcity. One is the scarcity-reduction route, in which one seeks to reduce the discrepancy in resources. The second is the control-restoration route, in which one aims to reestablish security and control in other domains. Under normal circumstances, when there is high recoverability from scarcity and an individual believes that scarcity can be improved by investing more time, money, and cognitive effort, the individual tends to prefer the scarcity-reduction route. Activities in this route consume an individual’s cognitive resources, thus affecting performance in other tasks. In contrast, when an individual believes that recoverability from scarcity is unlikely, they develop negative feelings and tend to compensate in other domains [29]. In the present study, participants were randomly assigned to a high- or low-scarcity group. For people who were assigned to the high scarcity group, the inadequate chances of assistance rendered in the word puzzle task were unmodifiable. Thus, recoverability from scarcity was unlikely. Individuals might not take the scarcity-reduction route, in which cognitive resources might be consumed, leaving less for the following social comparison task. Instead, they might take the control-restoration route to reestablish security and control in social comparison, such as emphasizing the desirability of their characteristics in social comparison. The results provided evidence for the self-management model of resource scarcity. Recoverability from scarcity was one of the factors that must be considered when we discuss the effect of sense of scarcity.

Interestingly, we also found that the above-average effect of individuals in negative traits was significantly greater than that in positive traits, which was consistent with the findings of a previous study in Chinese culture [15]. However, it was inconsistent with the results from Western culture, in which no such difference between positive and negative traits was found [18]. We speculated that, compared with Western culture, modesty is an important social norm in Chinese culture. Individuals in Chinese cultures might be as likely to have positive illusions about the self as individuals in Western cultures, but their positive illusions are expressed in different ways. Compared with self-enhancement motivation when evaluating positive traits, individuals had greater self-protection motivation when evaluating negative traits.

Although the causal relationship between a sense of scarcity and the above-average effect was confirmed in the present study, there were some limitations that made us interpret the findings with caution. Firstly, only college students were recruited as the participants, and most of them were female. The sample was, therefore, too homogenous to allow the generalizability of the study. Secondly, some control variables, such as level of self-esteem and gender differences, were not well-measured and counterbalanced in the study. Although a randomization method was used to assign participants to high- or low-scarcity groups and to offset personality and individual differences between the two groups, we cannot check its effectiveness. Moreover, previous studies have found that individuals with high levels of self-esteem had more positivity bias than those with low self-esteem [30,31]. Additionally, men also manifested greater positivity bias than women [32,33]. Thus, they might be likely to take different self-protection strategies to cope with threats under the circumstance of scarcity. It is worth examining how these personality and individual differences modulate the effect of scarcity on the above-average effect in a future study. Thirdly, previous studies have argued that a sense of scarcity could be from a lack of food, money, time, or resources [1,5]. Food and money are necessities to fulfill people’s basic needs. A lack of these necessary resources could generate a sense of scarcity. Meanwhile, some other feelings might also be activated, such as hunger, physical pain, and a sense of economic insecurity [7,8,34]. In the present study, a lack of chances for assistance in a cognitive task was used to activate one’s sense of scarcity. Individuals’ responses to the direct manipulation check question showed a sense of scarcity could be activated effectively in this way. However, it must be admitted that, compared with financial scarcity and food scarcity, a lack of chances for assistance in a cognitive task may not relate closely to real-world experiences of scarcity. Future studies could further confirm the effect of financial and food scarcity on the above-average effect to increase the ecological validity of this study. Subtle questions about one’s perceived scarcity could also be introduced to check the effectiveness of the scarcity manipulation. Lastly, the present study implied that under the circumstance of scarcity, the “above-average” response might be a self-protection strategy for people, serving an adaptive function and maintaining mental health. There are many expressions of positivity bias in humans. For example, during attributional interpersonal events, people with a sense of scarcity might attribute more positive and less negative events to themselves than to another person [15,20]. When imagining the future, they might overestimate positive events and underestimate negative events [35]. When receiving self-relevant social feedback, they might change their self-evaluations more based on desirable rather than undesirable feedback [11]. Possibly because of these enhanced positive illusions, people may believe that they can overcome hardships. It was thus worth studying whether and how people regulate themselves in these domains to cope with a sense of scarcity.

## 5. Conclusions

People with a high sense of scarcity judged their personality to be more desirable than those with a low sense of scarcity. That is, a sense of scarcity enhanced the above-average effect in social comparison. Importantly, it is worth noting that although the above-average effect may temporarily alleviate negative emotions and protect self-concept, it may not be conducive to coping with scarce situations in the long term. For example, when we receive negative social feedback, such as poor performance in games or academic tests, we emphasize our positive traits heuristically to alleviate negative emotions and maintain mental health. However, in the long term, the “above-average” response might enable people to avoid and neglect their weaknesses. Then, they may have no motivation to engage in self-reflection and develop their abilities. It is harmful for people to solve problems and achieve life success. Therefore, people need accurate knowledge about the self, even when it may be unflattering to one’s self-concept. A future study could focus on utilizing self-regulation to address the problem of scarcity in the long term. For example, previous studies have found that self-affirmations—prompting people to focus on their overall self-worth—can remind them of their psychosocial resources and intrinsic aspirations, foster an approach orientation to challenges, and help to clarify purpose in life [36,37,38]. Self-affirmation is considered an effective intervention tool in many domains, such as regulating negative emotions and preventing alcohol or smoking abuse [39,40,41]. Therefore, it might be helpful to acquire psychological resources to engage in goal-oriented behaviors rather than remaining stuck in a mindset of scarcity.

## Figures and Tables

**Figure 1 behavsci-13-00826-f001:**
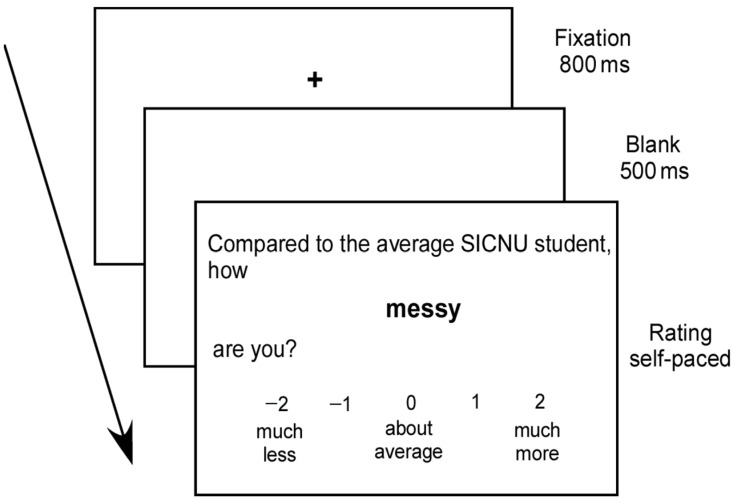
Stimuli and timing for the experimental procedure. The arrow indicated the sequence of the slide in a trial.

**Table 1 behavsci-13-00826-t001:** High- and low-scarcity participants’ ratings and RT (ms) on positive and negative personality items in social comparison task (M ± SE).

Sense of Scarcity	Positive		Negative	
	Ratings	RT	Ratings	RT
High	0.27 ± 0.06	2207.46 ± 136.41	0.88 ± 0.10	2387.11 ± 148.09
Low	0.19 ± 0.06	2120.49 ± 136.41	0.60 ± 0.10	2132.23 ± 148.09

## Data Availability

The data that support the findings of this study are openly available in OSF at https://osf.io/gr2xz (accessed on 26 January 2023).

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
