# Peer review of "A Sense of Scarcity Enhances the Above-Average Effect in Social Comparison"

_behavsci, 2023, doi:10.3390/bs13100826_

Round 1
Reviewer 1 Report
This is an interesting experimental study that shows that creating a sense of scarcity in participants affects how they subsequently view themselves, specifically, participants exposed to scarcity report more positive and less negative traits than participants not exposed to scarcity. The study has many strengths, such as the experimental design and the contribution to the literature on the psychological effects of scarcity. I think, however, that the manuscript can be improved in several areas:
- The authors describe the literature on the psychological consequences of scarcity and the literature on the above-average effect. However, the link between these two, and the resulting expectations the authors have for their own study, should be explained better. Why would the above-average effect occur more strongly among individuals who are in some way or another exposed to scarcity? Doesn't scarcity not also cause lowered self-esteem (see ref. #8)? The authors should argue for their speculation much more strongly.
- The methodology is lacking in detail in some places - for example, it only becomes clear in the results section that the authors have also measured reaction times. But it is not clear at which point these reaction times were measured - while participants were completing the puzzles, or when they were responding to the self-evaluation questions? And did authors have any expectations regarding these reaction times (e.g. did they expect longer or shorter response times for high or low scarcity participants? And why?)?
- Did the authors include any control variables? For example, were participants' self-evaluations or self-esteem measured before they started with the puzzle task?
- Were the self-evaluation questions randomized? Was there a difference visible for the first 50% of responses and the final 50% of responses? It would seem that the participants spend a relatively long time answering all these questions, and fatigue or disinterest may occur at some point.
- What was the composition of gender in your sample?
- Results section also needs more information: More details on the main effects and interaction effects need to be provided, i.e. the full model needs to be reported in the results section. Moreover, should there not be a MANOVA rather than 2 ANOVA analyses? What was the correlation between the negative & the positive self-evaluations?
- Did you investigate whether there were any gender differences? Magee & Upenieks (2019) report the following: "[...] self-esteem differs by gender due to a greater tendency for men to agree with positively worded self-statements, and a greater tendency for women to agree with negatively worded self-statements." --> this makes it relevant for the authors to at a minimum control for gender in their dependent variables.
- Discussion is more clear in explaining the effects of scarcity on self-evaluation (i.e. the 2 different ways of repairing the threat to the self); this might be a relevant explanation to include in your introduction as well to clarify the issues I stated above.
Sufficiently good
Author Response
Dear professor,
Many thanks for your enlightening comments and suggestions on our manuscript. We have now revised our paper. The modified parts were marked with blue-colored words in the revised manuscript.
For the first question, we’ve rewritten the introduction and explained more clearly about the resulting expectations (Lines 64-76).
For the second question, reaction times were measured in the social comparison task. We described it in the method section of the revised manuscript (127-129, 144-146). And we added some explanations about the expectations regarding the RT in the section of introduction (Lines 74-76 and 83-85).
For the third question, we did not measure self-esteem before we started with the puzzle task. And we just randomly assigned participants into a high or low scarcity group. But just like what you said, previous studies have found that people with high self-esteem would manifest more self-positivity bias. Future studies could further examine how self-esteem modulate the effect of sense of scarcity on the above-average effect. We added this information into the limitation and future directions of the study (Lines 291-300).
For the fourth question, the self-evaluation questions were randomized. We checked the difference between the first 50% of responses and the final 50% of responses. Results showed that there was no difference between them. These results were showed in the manipulation check section of the revised manuscript (Lines 198-204).
For the fifth and seventh questions, there were 69 participants in the present study. Of them, 58 females. We added this information in method section (Lines 164-165,183-184). The samples might be too homogenous to limit the generalizability of the study. And the gender difference could not be measured in the present study. We added these limitations and future directions in the section of discussion (Lines 289-300).
For the sixth question, consistent with the previous studies, we introduced positive and negative trait words to measure the above-average effect in social comparison [1, 2]. The positive self-evaluation could be used to measure the self-enhancement process of the above-average effect. While the negative could be used to measure the self-protection process of the above-average effect. They did not interact with each other. Thus, the valence of the trait words was designed as a within-subject factor. We used the general linear models and repeated measures ANOVA to examine the effect of sense of scarcity on the above-average effect. We added these details in the section of results (Lines 212-230).
- Beer, J.S. and B.L. Hughes, Neural systems of social comparison and the 'above-average' effect. NeuroImage, 2010. 49(3): p. 2671-2679.
- Hughes, B.L. and J.S. Beer, Protecting the Self: The Effect of Social-evaluative Threat on Neural Representations of Self. Journal of Cognitive Neuroscience, 2013. 25(4): p. 613-622.
Reviewer 2 Report
This is a well-done study providing valuable evidence that scarcity heightens positive self-evaluation. Specifically, the paper investigates an interesting and novel research question about how a sense of scarcity affects self-evaluation and the above-average effect. This contributes new knowledge to the literature on scarcity and social comparison.
The paper has various merits, namely:
- The experiment is well-designed with manipulation of a high vs low sense of scarcity and measurement of the above-average effect. The sample size seems adequately powered based on a priori calculations, although the sample is not gender balanced. However, this might have affected the results and should be noted in the discussion/limitations.
- The results support the main hypothesis that a high sense of scarcity enhances the above-average effect compared to a low sense of scarcity. This results, which is intuitively appealing, provides evidence that scarcity leads to positive illusions about the self, likely as a self-protection mechanism. It would be useful that in the discussion, the authors connect the interpretation of the results about the self-management self-protection effects to the relevant literature more thoroughly and systematically.
- The paper is clearly written for the most part, with a logical flow from introduction to discussion. The introduction effectively reviews relevant literature and therefore provides an appropriate rationale for the study.
However, in addition to the suggestions already made above, the paper in its current version also has weaknesses that should be addressed in a revised version.
From a logical point of view, I would like the link between scarcity and self-protection motives driving greater above-average effect to be more clearly explained in the introduction.
Moreover, as already pointed out above, the discussion should relate the findings better to the self-management model of resource scarcity described earlier. How do the results support the control-restoration response route?
From a methodological point of view, the manipulation check question is very direct. It’s good to know that it was effective, but it would be useful if the authors address the issue of whether more subtle questions about perceived scarcity might have been better also as a direction for future research.
As already remarked, the sample is well-powered but also, as it is typical of most of the literature, quite homogenous (college students), obviously limiting generalizability. This should be clearly noted in the discussion. Most of the psychology that we know about from experimental studies is the psychology of college students…
The manipulation is based on chances for assistance in a cognitive task, which may not relate closely to real-world experiences of scarcity. This point should be clearly addressed in the discussion and, if possible, the methodological choice made should be better argued and defended by the authors. Using financial scarcity may increase ecological validity, and it could be useful to note this point in the discussion.
From a statistical point of view, effect sizes for the main analyses should be reported, it would add helpful context.
Finally, in terms of expositional clarity, I have two remarks. First,he abstract does not report the key finding about scarcity enhancing above-average effect, but just hints at a self-judgment of better desirability. This part should be revised for more clarity.
Moreover, the results section is very compact and difficult to read. The authors should consider adding some sentence to better highlight the meaning of the single results.
Overall, the study makes a useful contribution in demonstrating the causal effect of scarcity on greater positive illusions. This adds to knowledge of how people adapt to challenging conditions and is a valuable scientific contribution.
However, the authors could more clearly highlight the study's implications, such as how scarcity may temporarily boost self-views but undermine long-term coping. Moreover, they should elaborate a bit more about the study’s implications in the conclusions.
Quality of English is good but a check of typos/minor inconsistencies is advised.
Author Response
Dear professor,
Many thanks for your enlightening comments and suggestions on our manuscript. We have now revised our paper. The modified parts were marked with blue-colored words in the revised manuscript.
For the first question, we’ve rewritten the introduction and explained more clearly about the resulting expectations (Lines 64-76).
For the second question, we’ve rewritten the discussion section to explain our design and findings (245-276).
For the third question, about the manipulation check questions, we added the subtle way as a future direction (312-314).
For the fourth question, as you said, our samples were all college students, it might be too homogenous to limit the generalizability of the study. We added it into the limitations and future directions section of discussion (Lines 289-291).
For the fifth question, we discussed the reason why we manipulated sense of scarcity by budgeting one’s chances for assistance in a cognitive task. For example, previous studies have argued that a sense of scarcity could be from a lack of food, money, time, or resources [1, 5]. Food and money, which are a necessity to fulfill the people's basic needs. Lack of these necessity resources could generate a sense of scarcity. Meanwhile, some other feelings might be also activated, such as hungry, physical pain, and sense of economic insecurity. In the present study, lack of chances for assistance in a cognitive task was used to activate one’s sense of scarcity. Individuals’ responses to the direct manipulation check question showed a sense of scarcity could be activated effectively by this way. But it must be admitted that, compared with financial scarcity and food scarcity, lack of chances for assistance in a cognitive task may not relate closely to real-world experiences of scarcity. Future study could further confirm the effect of financial and food scarcity on above-average effect to increase ecological validity of the study (Lines 301-311).
For the sixth question, effect sizes for the main analyses were reported in the revised manuscript (Lines 193-197).
For the seventh question, the abstract was rewritten and the key findings about scarcity enhancing above-average effect were reported clearly in the revised manuscript (Lines 16-20).
For the eighth question, we added some sentences to explain the meaning of the results (Lines 187-189, 203-204, 210-211, 220-223, 229-230).
Round 2
Reviewer 1 Report
Thank you for responding to my (and the other reviewer's) comments and for revising the manuscript so thoroughly. I am very happy with your changes and I think you have delivered a very nice piece of work.
Some minor grammatical issues in the new additions to the text.